# Expanding the Scope of Orthogonal Translation with Pyrrolysyl-tRNA Synthetases Dedicated to Aromatic Amino Acids

**DOI:** 10.3390/molecules25194418

**Published:** 2020-09-25

**Authors:** Hsueh-Wei Tseng, Tobias Baumann, Huan Sun, Yane-Shih Wang, Zoya Ignatova, Nediljko Budisa

**Affiliations:** 1Institut für Chemie, Technische Universität Berlin, Müller-Breslau-Straße 10, 10623 Berlin, Germany; justin3926@hotmail.com (H.-W.T.); tobias.baumann@tu-berlin.de (T.B.); huansunluck@gmail.com (H.S.); 2Institute of Biological Chemistry, Academia Sinica, Taipei 116, Taiwan; yaneshihwang@gate.sinica.edu.tw; 3Institute of Biochemical Sciences, National Taiwan University, Taipei 116, Taiwan; 4Institute of Biochemistry and Molecular Biology, University of Hamburg, 20146 Hamburg, Germany; zoya.ignatova@uni-hamburg.de; 5Department of Chemistry, University of Manitoba, Winnipeg, MB R3T 2N2, Canada

**Keywords:** aromatic amino acid analogs, site-specific incorporation, genetic code expansion, *Methanosarcina mazei* pyrrolysyl-tRNA synthetase (*Mm*PylRS), non-canonical amino acids (ncAAs), orthogonal pairs, ribosomal translation

## Abstract

In protein engineering and synthetic biology, *Methanosarcina mazei* pyrrolysyl-tRNA synthetase (*Mm*PylRS), with its cognate tRNA^Pyl^, is one of the most popular tools for site-specific incorporation of non-canonical amino acids (ncAAs). Numerous orthogonal pairs based on engineered *Mm*PylRS variants have been developed during the last decade, enabling a substantial genetic code expansion, mainly with aliphatic pyrrolysine analogs. However, comparatively less progress has been made to expand the substrate range of *Mm*PylRS towards aromatic amino acid residues. Therefore, we set to further expand the substrate scope of orthogonal translation by a semi-rational approach; redesigning the *Mm*PylRS efficiency. Based on the randomization of residues from the binding pocket and tRNA binding domain, we identify three positions (V401, W417 and S193) crucial for ncAA specificity and enzyme activity. Their systematic mutagenesis enabled us to generate *Mm*PylRS variants dedicated to tryptophan (such as β-(1-Azulenyl)-l-alanine or 1-methyl-l-tryptophan) and tyrosine (mainly halogenated) analogs. Moreover, our strategy also significantly improves the orthogonal translation efficiency with the previously activated analog 3-benzothienyl-l-alanine. Our study revealed the engineering of both first shell and distant residues to modify substrate specificity as an important strategy to further expand our ability to discover and recruit new ncAAs for orthogonal translation

## 1. Introduction

Protein translation is a process by which a protein sequence is being built up according to the information provided in the messenger RNA. In nature, there is a limited set of canonical amino acids (cAAs) that can be utilized as building blocks in this process. Conversely, non-canonical amino acids (ncAAs) can be incorporated into recombinant proteins using a number of in vivo and in vitro methods [1,2]. These non-natural insertions provide useful modifications that might take place on the protein backbone or on the amino acids’ side chains. In nature, protein backbone modifications are the result of various post-translational modifications (PTMs), whereas in the frame of reprogrammed protein translation, the easiest way to in vivo modification is to use proline analogs [3,4,5,6]. Other protein backbone modifications are typically produced using in vitro ribosomal translation systems. For example, β-, or γ-amino acids can be inserted into target polypeptide sequences by using engineered in vitro translation systems [1]. Notably, α-hydroxy acids can also be incorporated into proteins, for example in response to in-frame amber codons via the action of specific engineered systems [7]. However, the vast majority of ncAAs delivers modifications of protein side chains with unique functionalities. A typical example in nature is the rare 22nd proteinogenic amino acid pyrrolysine (Pyl), which appears in very few proteomes of methanogenic microorganisms [2].

Pyrrolysyl–tRNA synthetase (PylRS) is a particularly powerful enzyme that enables the incorporation of various amino acid substrates into a protein. Its original natural substrate is the rare proteinogenic amino acid pyrrolysine (Pyl), a lysine analog with a 4-methyl-pyrroline-5-carboxylate ring attached to the lysine side chain [8,9,10]. It is recognized and enzymatically charged onto tRNA^Pyl^, forming a complex Pyl–tRNA^Pyl^ via the action of PylRS, which is a class II aminoacyl–tRNA synthetase (aaRS) [11]. The anticodon of the cognate tRNA is CUA, providing a natural PylRS:tRNA^Pyl^ orthogonal pair. This pair fulfils all requirements for a natural orthogonal translation system (OTS) capable of in-frame stop codon suppression (SCS) and subsequent site-specific ribosomal incorporation of an ncAA substrate into a recombinant target protein [12].

For a new ncAA to become amenable for genetic encoding, a specific aaRS variant is required, whereas “polyspecific” variants recognize several substrates (e.g., 3-bromo-, 3-iodo-, and 3-trifluoromethyl-l-phenylalanine) [13]. Commonly, the selection of molecular machineries capable of ncAA incorporation is done via generation of gene libraries of a chosen aminoacyl-tRNA synthetase. Two natural archaeal systems, the tyrosine pair of *Methanocaldococcus jannaschii* (*Mj*TyrRS:tRNA^Tyr^) and the pyrrolysine pair of the *Methanosarcina* species (PylRS:tRNA^Pyl^) were rationally constructed and used for the selection of new OTSs in *Escherichia coli* (*E. coli*) [14,15,16,17]. To date, these synthetase scaffolds (*Mj*TyrRS and PylRS) are predominantly used and enable site-specific incorporation of diverse ncAAs into proteins in vivo. Orthogonal pairs (o-pairs) based on both the *Mj*TyrRS and *Mm*/*Mb*PylRS scaffolds can have advantages and disadvantages in different experimental settings. For each aaRS variant, it is crucial that cAAs are excluded from activation and thus not charged onto the suppressor tRNA. In bacterial host cells, *Mj*TyrRS often show a higher performance (e.g., yield of purified ncAA-modified target protein) compared to PylRS-based systems. Commonly, orthogonal translation with PylRS-derived o-pairs leads to low yields of target protein, since this enzyme generally has a low catalytic efficiency. While o-pairs based on *Mj*TyrRS are commonly efficient enough to allow a higher number of in-frame stop codons that can be suppressed, this is not the case with orthogonal translation based on PylRS.

The PylRS system particularly covers a wide range of applications for genetically encoding ncAAs with bio-orthogonality in both prokaryotes and eukaryotes. Thus, tRNA^Pyl^ and PylRS represent a so-called orthogonal pair for the corresponding host cells. Notably, the catalytic core containing the ncAA substrate binding region is predominantly built of hydrophobic residues, where mutation Y384F can increase the activity of PylRS [18]. The corresponding protein crystal structure showed that the binding pocket was essentially unchanged, but the mutation enlarged the substrate range of the enzyme [19]. The evolutionary origin and structural resemblance to PheRS explain why PylRS and specific Y384 mutants exhibit better activity towards phenylalanine or tryptophan (as well as their analogs) [20].

In this study, we combined the accumulated findings of various PylRS engineering and improvement reports in order to expand the scope of *Mm*PylRS toward novel aromatic ncAA analogs (Figure 1) amenable to be used in both prokaryotic and eukaryotic hosts. These moieties are often valuable probes for studying non-covalent interactions, hydrophobicity, and polarity effects in protein structure, folding and activity. We constructed randomly and rationally designed *Mm*PylRS gene libraries to explore the diversity of ncAA incorporation [21,22]. Site-saturation mutagenesis (SSM) via NNK codons (N: A/T/C/G; K:G/T) was used to specifically randomize chosen positions of the aaRS [23]. Our strategy is further based on an analysis of the available high-resolution molecular structure of *Mm*PylRS, as well as earlier reports showing that mutagenesis of the crucial positions N346 and C348 (Figure 2) is feasible to expand the substrate range of this enzyme [24,25]. We make a step further and include other residues in our mutagenesis schemes, importantly including second and higher shell residues, which are expected to be important for ncAA substrate binding and recognition, but not in direct contact. Our approach proves to be successful as we were able to design novel enzyme variants that specifically recognize a set of aromatic analogs. In particular, a much higher ribosomal incorporation efficiency can be achieved for 3-benzothienyl-l-alanine (Bta) relative to a previous study [26].

Joining current efforts, our main goal is to evolve *Mm*PylRS variants dedicated to various tyrosine and tryptophan analogs that have so far only been incorporated into proteins by *Mj*TyrRS [16]. We succeeded to integrate 10 different aromatic analogs into this OTS using semi-rationally engineered *Mm*PylRS variants.

## 2. Results

### 2.1. Engineering MmPylRS Variants for Aromatic Non-Canonical Amino Acids

#### 2.1.1. *Mm*PylRS Mutagenesis Strategy and Screening towards Tryptophan and Tyrosine Analogs

Initial *Mm*PylRS gene libraries were constructed by mutating two key aaRS positions: N346 and C348 (Figure 2). Rational aaRS design aimed to provide specific enzyme–substrate interactions that enable binding and activation of the corresponding ncAAs. Accordingly, active site residue N346 was mutated to A, G, and S, whereas C348 was mutated to Q, N, and M. In previous studies, the structures of two PylRS variants (N346G/C348Q and N346S/C348Q) were modeled, which revealed crucial interactions between these aaRS side chains and substrates in the form of aromatic amino acids [18,24,25,27,28]. Due to the increased bulkiness of the ncAA analogs, enlarging the enzyme binding pocket may improve the access and recognition of the aromatic moieties, allowing charging onto tRNA^Pyl^ and ribosomal incorporation. Next, based on the PylRS structure and activity data collected from a large number of enzyme mutants, residues V401 and W417 (Figure 2) were targeted. These were expected to define ncAA specificity towards hydrophobic or polar amino acids [10,29]. Moreover, positions L305, Y306, and L309 shape the aaRS active site pocket. These five positions (L305, Y306, L309, V401, and W417) were chosen for full randomization as they have direct influence on enzyme activity (Figure 2). The PylRS Y384F mutation, which conferred an increased aminoacylation rate in previous studies, was introduced into the starting PylRS sequence and remained fixed in all mutants [18,20]. Novel and semi-rationally designed *Mm*PylRS gene libraries capable of activating aromatic ncAAs were successfully constructed. To detect the ribosomal incorporation of the chosen Trp and Tyr analogs, a reporter construct of super–folder green fluorescent protein (sfGFP) was used. The sfGFP gene contained an in-frame amber stop codon at the position downstream of the first methionine encoding codon (sfGFP–R2TAG) and a C-terminal His-tag. In this way, the sfGFP fluorescence signal of intact cells can be used as readout for the in vivo suppression of the in-frame amber stop codon, with fluorescence intensity and full-length protein yields proportional to suppression efficiency [23].

Next, a set of aromatic ncAAs were selected for testing (Figure 1). These substrates were supplied to *E. coli* BL21(DE3) cell cultures co-expressing the genes of designed *Mm*PylRS variants with cognate tRNA^Pyl^ and the sfGFP–R2TAG reporter. We utilized 96-well plate setups with *Mm*PylRS variants and ncAAs screening protocols similar to those established by the Söll lab to analyze changes in the aaRS efficiency and substrate specificity [13,20,30]. Cells were grown in 96-well plates filled with chemically defined medium separately supplemented with all 20 cAAs, which leads to an observable fluorescence background, and individual target ncAAs (Appendix A). Owing to the established reporter methodology, the efficiency of ncAA charging onto the amber suppressor tRNA^Pyl^ in vivo is directly correlated with the intensity of the fluorescence signal. Similar to recent reports, these screenings revealed fluorescence of sfGFP upon incorporation of aromatic Tyr or Trp analogs mediated by the individual *Mm*PylRS variants [31,32]. Herein, the different enzyme variants are named according to the introduced mutations, e.g., *Mm*PylRS–SMG corresponds to the aaRS mutation set N346S/C348M/V401G. Mutations in the individual enzyme variants are listed in Appendix A. Activity screenings revealed fluorescence signals significantly above background with *Mm*PylRS–SMG in the presence of Bta and 1-NaA (Figure 3A), whereas *Mm*PylRS–GML (mutation set N346G/C348M/W417L) was capable of activating 3-ClY and 3-BrY (Figure 3B). These results demonstrated that the *Mm*PylRS variants with V401 mutations were amenable to Trp analogs, while aaRS variants containing W417 mutations were capable of Tyr analog activation and tRNA charging (Appendix A).

#### 2.1.2. Intact Protein Mass Analysis of Aromatic ncAA Incorporation

To validate the results from the screening assays, protein mass spectra were acquired. NcAA-modified sfGFP reporter protein variants were produced in bacterial cells via the action of the *Mm*PylRS variants, purified and analyzed in electrospray LC–MS (Q–TOF, quadrupole time-of-flight mass spectrometry). Intentionally, standard T7 expression *E. coli* cells were used, but we want to remark that sophisticated strains for improved amber suppression have been developed [33]. The presence of aromatic ncAAs generates characteristic mass shifts in the intact protein mass spectra. As expected from previous results shown in Figure 3, the Trp derivatives Bta and 1-NaA were successfully incorporated by co-expression of *Mm*PylRS–SMG, while *Mm*PylRS–GML mediates efficient incorporation of the Tyr derivatives 3-ClY, 3-BrY, and furthermore 3-IodY (Figure 4). In this way, the mass analysis corroborates the small-scale fluorescence assay results.

### 2.2. Improvement of Amber Suppression Efficiency

#### 2.2.1. Bta Incorporation Efficiency Compared to Other *Mm*PylRS Variants

Although the results showed the *Mm*PylRS variants successfully activating aromatic ncAAs, the efficiency of PylRS-based stop codon suppression in bacterial cells remained relatively low. This can be seen from weak fluorescence signals from the reporter proteins containing ncAAs, as reported in [13,34]. Considering the poor enzyme kinetics of many engineered PylRS systems, we chose to further optimize the enzyme variants. Bta is a close structural analog of Trp, and it was incorporated by the *Mm*PylRS variant dubbed BtaRS carrying the two active site mutations N346G and C348Q [26]. We carried out a site-directed mutagenesis towards an improved Bta incorporation and found that mutation of position V401 to glycine led to a multifold improvement in Bta incorporation efficiency (Appendix A). Several other strategies to achieve better ncAA incorporation have recently been developed. For example, the archaeal *Mm*PylRS gene can be codon-optimized to enhance the recombinant synthetase gene expression and its overall efficiency within the bacterial host cell. As reported earlier for *Mb*PylRS, this can increase the intracellular amount of soluble and functional synthetase, which in turn increases full-length target protein yields [35]. When compared with the natural archaeal gene sequence as used in previous works, the codon-optimized *Mm*PylRS gene led to an increased ribosomal Bta incorporation efficiency (Appendix A)

#### 2.2.2. Enhancing the Efficiency of Ribosomal Bta Incorporation and Target Protein Yields

Previous studies have documented aaRS mutations, especially in the N-terminal portion of PylRS, which led to improved synthetase activity. Moreover, a chimeric pyrrolysyl–tRNA synthetase variant (*ch*PylRS, comprising the N-terminal aaRS part of *Mb*PylRS and the C-terminal part of *Mm*PylRS) was reported to benefit from the mutations V31I, T56P, H62Y, and A100E (IPYE), which were found by continuous directed evolution [34]. We compiled previously found positions in PylRS that can be beneficial for aaRS activity in the recombinant setting. The resulting set of sites (T13, V31, I36, T56, R61, H62, H63, A100, and S193) was chosen for mutagenesis studies to further improve Bta incorporation, monitored as before using the sfGFP reporter (Appendix A) [36,37]. From this dataset of rational PylRS mutagenesis and activity screening, it became apparent that considerable improvements arise upon mutation of aaRS position S193. Thus, we chose to pursue a full site saturation mutagenesis, generating and testing all 19 possible enzyme variants (Appendix A). For the efficiency of Bta incorporation, replacement of S193 by positively charged (Lys, Arg, and His), amidic (Asn and Gln), or small side chain (Ala and Gly) amino acids proved beneficial. However, when S193 was replaced with negatively charged (Asp and Glu) or aromatic (Phe, Tyr, and Trp) amino acids, the efficiency of stop codon suppression dropped (Appendix A). The aaRS protein structure solved in previous studies (see Appendix B) indicates that position S193 is located in the C-terminal domain of *Mm*PylRS but spatially close to the N-terminal domain, and is thus associated with tRNA charging activity [36,38,39]. Based on the activity profiles, PylRS variant SMG was combined with the mutation sets IV (T13I and I36V) [37], IPYE (V31I, T56P, H62Y, and A100E) [34], and KYR (R61K, H63Y, and S193R) [36,40], respectively, to improve ribosomal Bta incorporation in vivo (Figure 5A). As a result, *Mm*PylRS–KYR_SMG, which includes the potent S193R mutation, led to higher sfGFP reporter fluorescence. 

Therefore, we propose that mutations of S193 (especially to positively charged residues) provide a good strategy to enhance the ncAA incorporation efficiency of the PylRS system. For the obtained data sets, we chose to continue creating additional PylRS mutation sets and combinations thereof. Notably, the *Mm*PylRS–13IPYER_SMG variant (containing mutations T13I, T56P, H62Y, A100E, S193R, N346S, C348M, and V401G) yielded the highest Bta-dependent reporter fluorescence intensity (Figure 5B). At the same time, the background of stop codon suppression (reporter signal in the absence of Bta supplementation) is not increased. This indicates that the efficiency and specificity of *Mm*PylRS–13IPYER_SMG in vivo are greatly enhanced by introducing these combined mutations, yielding more ncAA-modified target protein for the same amount of ncAA supplied.

Target protein production and purification were performed to further validate the increased Bta incorporation efficiency by the newly generated *Mm*PylRS variants. As in previous works, we fused the sfGFP (R2TAG) reporter with a His- and a small ubiquitin-like modifier (SUMO) tag at the N-terminus to enable detection and purification of the truncation product generated by translation termination at the amber position [23]. Protein production was performed in the presence of Bta and *Mm*PylRS variant co-expression. In this setup, aaRS variants enabling high Bta incorporation efficiencies should predominantly lead to the production of full-length target protein, whereas those with poor efficiency should result in lower amounts of full-length protein and increased amounts of truncation product. Four *Mm*PylRS constructs were chosen for evaluation, namely *Mm*PylRS–SMG, codon-optimized *Mm*PylRS–coSMG, *Mm*PylRS–KYR_SMG, and *Mm*PylRS–13IPYER_SMG. The target proteins were purified through the N-terminal His-tag and yields were quantified. Remarkably, *Mm*PylRS constructs with the S193R mutation (*Mm*PylRS–KYR_SMG and *Mm*PylRS–13IPYER_SMG) ribosomally expressed mostly full-length fusion protein with only few truncation products, as evidenced by SDS–PAGE (Appendix A). Obtained protein yields were calculated per liter of bacterial culture in M9 minimal medium and measured by absorbance at 488 nm, which originates from the sfGFP chromophore. Yields of full-length SUMO–sfGFP were 6 mg for *Mm*PylRS–SMG, 11 mg for *Mm*PylRS–coSMG, 12 mg for *Mm*PylRS–KYR_SMG, and 14 mg for *Mm*PylRS–13IPYER_SMG, nicely demonstrating the effect of the efficiency-improving secondary aaRS mutations.

### 2.3. Improvement of Enzyme Efficiency Toward Other ncAAs

Bta incorporation efficiency was strongly increased through our combined *Mm*PylRS engineering strategies. Next, we set out from our semi-rationally designed aaRS scaffold to test if more and novel ncAAs can be activated. Indeed, some new ncAAs were incorporated into the sfGFP reporter upon co-expression of the improved *Mm*PylRS variants. Albeit with relatively low efficiency, β-(1-azulenyl)-l-alanine (AzAla) was incorporated in case of the MmPylRS–SMG mutants with the mutation sets KYR and W417 amino acid substitutions (W417L, W417K, or W417I). Moreover, 4-benzoyl-l-phenylalanine (Bpa) was incorporated by *Mm*PylRS–SMI (N346S, C348M, and W417I) and *Mm*PylRS–SML (N346S, C348M, and W417L) variants bearing the KYR modification (Figure 6B). To validate the fidelity of ncAA incorporation, the purified target proteins were analyzed by ESI-MS. The results were consistent with the data from fluorescence screening and showed that both AzAla and Bpa were successfully incorporated into the target protein (Appendix A). Moreover, mass spectra of the intact proteins revealed successful installation of the ncAAs 1-MeW (*Mm*PylRS–KYR_SMG) and 3-MeY (*Mm*PylRS–KYR_SMGI) (Appendix A).

### 2.4. Evaluation of Amber Suppression Efficiency and Other Growth Media

To further evaluate the efficiency of the created PylRS variants, wild-type sfGFP (wt–sfGFP) without an in-frame amber stop codon was included in reporter assays. Based on previous fluorescence screens, highly active aaRS constructs were selected for analysis. As seen before, the site-specific incorporation of Bta into the target protein is most efficient when mediated by *Mm*PylRS–KYR_SMG (Appendix A). Comparing the fluorescence intensities, this OTS reaches a suppression efficiency of 50% relative to the wt–sfGFP control under identical conditions. For an OTS based on PylRS, this represents high activity and target protein production [34]. Moreover, further controls were included at this stage. When only tRNA^Pyl^ (but no PylRS enzyme) was co-expressed, slightly elevated but still negligible background levels of amber suppression were observed. Under identical conditions, these were not elevated when the aaRS was also co-expressed but no ncAA supplied (Appendix A).

As often conducted for PylRS-based systems, our OTS activity screening and characterization were performed in chemically defined minimal media [28,41]. In order to test the robustness and versatility of the ncAA incorporation systems, target protein production was performed in a chemically-defined auto-induction medium developed for ncAA incorporation [42] and further in commonly used complex rich media. As above, wt–sfGFP was included as reference for recombinant protein production in the absence of amber suppression. Again, orthogonal translation with *Mm*PylRS–KYR_SMG displayed the highest Bta incorporation efficiency, reaching maximum yields when conducted in LB medium (Appendix A). Successful ncAA incorporation into the target protein was validated via ESI-MS (Appendix A). Commonly used for recombinant protein production (for example for protein crystallography), TB medium also proved suitable for the two top performing systems based on *Mm*PylRS–KYR_SMG and *Mm*PylRS–13IPYER_SMG, respectively.

## 3. Discussion

Nowadays, numerous ncAAs can be site-specifically incorporated into proteins of interest by expanding the genetic code. One of the major applications of OTSs is the possibility to mimic post-translational modifications (PTMs) in a precisely controlled manner [43]. In contrast, other traditional methods of protein modifications usually lead to non-specific labeling, yielding a heterogeneous and complex protein mixture [44]. Indeed, the microheterogeneity of proteins upon labelling is an important issue when attempting specific PTMs with desired chemical modifications, both natural and non-natural. Consequently, selective methods that allow absolute control of the position of the reactive handle within a protein are required [45].

Despite the widespread use of various PylRS:tRNA^Pyl^ pairs in orthogonal translation, the substrate range of the engineered aaRS variants, stemming from its natural one, is mostly restricted to lysine analogs with long, flexible side chains [10]. It is desirable to expand the substrate range of these systems towards amino acids with aromatic side chain derivatives such as Trp. For example, in many protein–protein interactions, Trp residues are essential [46,47]. However, their role is difficult to investigate via spectroscopy because their absorption and fluorescence signals are more or less indistinguishable from other non-essential Trp residues present in the protein [48]. In this context, Trp-like ncAAs are distinct chromophores offering unique advantages of intrinsic probes, especially in complex biological settings such as mammalian cells, tissues, and even whole animals. Furthermore, such ncAAs are likely a good tool for investigating protein–protein or protein–nucleic acid interactions due to their unique spectroscopic signature [49].

For installation of such moieties, widely used orthogonal *Mm*PylRS variants designed by evolutionarily or rationally based methods fit perfectly into this scheme. However, the low ncAA incorporation efficiency of PylRS in bacterial systems limits its utility for OTS development [34]. This is often mitigated by supplying higher amounts of chemically synthesized ncAA to the bacterial growth medium. For that reason, we present here a potentially useful approach to expand the substrate range and activity of *Mm*PylRS towards Trp and Tyr analogs, employing an enhanced ncAA set in microtiter plate screening, which quickly identifies novel substrates enabled by rational or random mutagenesis. Our approach has the following advantages over previously reported systems: (i) PylRS is compatible with both eukaryotic and bacterial expression systems, (ii) it enables the translation of spectroscopically valuable aromatic ncAAs (such as AzAla) and (iii) significantly improves the target protein yields of related OTSs for the same amount of ncAA supplied to the host cells.

In our study, mutations at positions V401 and W417 of *Mm*PylRS are expected to reshape the aaRS binding pocket, allowing the synthetase to activate the bulkier Trp and Tyr analogs. Based on the structures of wild-type and a mutant *Mm*PylRS, the binding pocket of the aaRS includes four critical residues (N346, C348, V401, and W417) [26]. Rational mutagenesis of these residues to smaller side chains might enlarge the binding pocket to accommodate larger ncAA substrates (Figure 2). In particular, N346 and C348 were suggested to act like substrate gating residues in earlier studies and their rational manipulation would make it possible to expand the substrate repertoire of *Mm*PylRS. The Bta-dedicated engineered enzyme BtaRS created by double mutations (N346G/C348Q) clearly supports this notion [26]. We engineered an alternative OTS with higher efficiency of Bta incorporation by mutation set N346S/C348M/V401G. This shows that based on the same aaRS scaffold, there can be multiple solutions to recognize and activate a given ncAA substrate. From the aaRS mutagenesis and screening results, it is evident that the subtle interactions between ncAA substrate and the enzyme can benefit from multiple mutations, which establish and fine-tune selectivity and activity.

Following the same strategy, aaRS variant *Mm*PylRS–GML was constructed, which efficiently activates halogen-modified Tyr analogs, leading to their site-specific ribosomal incorporation. Upon introduction of a second halogen moiety (e.g., 3,5-diClY and 3,5-diBrY), however, ncAA incorporation signals drop, which may indicate that, once again, the size and shape of the aaRS active site hits limits for the accommodation of more bulky ncAA substrates.

Previous studies reported that besides residues in the vicinity of the active site pocket, also N-terminal mutations within the PylRS sequence or can improve the efficiency of ncAA incorporation [34]. For instance, for the relatively weak starting level of AzAla incorporation, the impact of the activity-enhancing mutation set becomes evident. In this context, it is particularly important to note that the mutation of *Mm*PylRS residue S193 proved to enhance the Bta incorporation efficiency. This residue is located at the C-terminal domain of the aaRS and takes part in tRNA binding [40]. Thus, mutations of S193 may be beneficial due to the alteration of interaction between the synthetase and tRNA, and not the actual activation of the amino acid substrate. Consequently, S193 mutagenesis results may be transferrable to several previously developed PylRS-based systems.

Site-specifically installed, the use of Trp analogs is not only an interesting approach to modify protein spectral properties. Such analogs are also excellent tools for biophysical studies of proteins as non-invasive probes, capable of minimizing potential structural perturbation of the protein scaffold. A subtle exchange on aromatic side chains (e.g., -H->F, or -CH3), also known as “atomic mutations”, are especially useful for performing various fine-tuned biophysical studies [50,51]. For example, both Bta with its sulfur heteroatom and AzAla with its unique blue fluorescence emission and vibrational properties are interesting biophysical probes (e.g., exhibit a large dipole moment, a narrow energy gap between the HOMO and LUMO, and abnormal fluorescence) [52,53,54]. They would allow for observing the electronic vibration and transfer, which are particularly relevant to all aspects of protein structure, activity, and functions with charge stabilization [55]. In addition, both ncAAs are interesting biomedical agents and bioactive peptide building blocks [56,57,58].

Finally, *Mm*PylRS-based systems for Trp analogs have great potential in various omics setups. For example, Sakamoto and co-workers used several engineered PylRS variants for a proteome-wide reassignment of rare-sense codons with ncAA analogs [36]. Such studies could also be widened to cellular proteins and enzymes that bind DNA, such as molecular switches, restriction and repair enzymes, gyrases or nucleosomal and nucleic acids-associated proteins. It should not be difficult to imagine how repressor or activator proteins equipped with suitable Trp analogs can either block the access of enzymes such as polymerases to DNA or enhance their binding to DNA—opening up completely new avenues in investigating complex biological phenomena and processes [59].

## 4. Materials and Methods

### 4.1. Canonical and Non-Canonical Amino Acids

A total of 20 canonical amino acids were purchased from Sigma. Non-canonical amino acids were from Bachem, abcr GmbH, Alfa Aesar, Merck, Biosynth, TCI, and Abchem (see Appendix A).

### 4.2. Site-Directed Mutagenesis, Strains, and Screening Library Construction

*Mm*PylRS gene engineering was conducted via PCR using mutagenic primers with NNK (N = A, T, G, or C; K = G or T) randomizations at designated positions (S193, V401, and W417). Primers were synthesized by Merck, Darmstadt, Germany [28,60]. Primer sequences are listed in the Appendix A (Note S2). The pBU16 plasmid containing pyrrolysyl–tRNA_CUA_ driven by glnS’ promoter was digested with NdeI and KpnI [60]. The digested PCR products and linearized plasmid were ligated using T4 DNA ligase, and transformed into *E. coli* strain Top10. The gene of sfGFP containing an in-frame amber codon (sfGFP–R2TAG) used as a reporter for amber suppression was cloned into plasmid vector pET-28. For intact cell fluorescence experiments, an N-terminal His-tag fused with SUMO–sfGFP based on reporter constructs carrying an in-frame amber codon at position R2 was cloned in pET-28, as reported earlier [23].

### 4.3. Screening aaRS Variants for Activity

To screen the *Mm*PylRS variants for activity towards Trp and Tyr analogs, the two plasmids, pBU16–*MmPylRS–*${\mathrm{tRNA}}_{\mathrm{CUA}}^{\mathrm{Pyl}}$ and pET-28–*sfGFP_R2TAG*, were co-transformed into *E. coli* strain BL21(DE3) [60]. Single colonies were chosen from the plate and pre-cultured in 1 mL of LB medium supplemented with Amp (100 μg/mL) and Kan (100 μg/mL) at 37 °C overnight. Then, the pre-cultured cells were transferred (ratio 1:500) to 20 mL fresh LB medium and incubated at 37 °C. When the optical density (OD_600_) of the cultures reached 0.50–0.8, cells were centrifuged and resuspended twice in M9 minimal medium (M9 salts (7.52 g/L Na_2_HPO_4_·2H_2_O, 3 g/L KH_2_PO_4_, 0.5 g/L NaCl, and 0.5 g/L NH_4_Cl), 0.4% glucose, 1 mM MgSO_4_, 1 mg/mL thiamine, 1 mg/mL biotin, and 0.3 mM CaCl_2_). The cells were cultured in M9 minimal medium (with 100 μg/mL Amp, 100 μg/mL Kan, and 1 mM isopropyl-β-D-thiogalactoside (IPTG)) and distributed (200 μL) in a 96-well black µ-plate (Ibidi, Martinsried, Germany) containing different ncAAs at a final concentration of 1 mM in each well. The plates were incubated at 37 °C with shaking for 18 h. Fluorescence intensities and OD_600_ were monitored by an Infinite M200 microtiter plate reader (Tecan, Männedorf, Switzerland), the former via bottom reading (excitation wavelength of 481 nm, emission wavelength of 511 nm, and a manual gain of 85) every 10 min. The screening library system contains four wells for the background signals (control), 20 cAAs, and 50 ncAAs (full details of arrangement are in Appendix A).

### 4.4. Protein Production and Purification

To produce sfGFP with site-specific ncAA incorporation, plasmid vectors pBU16–*MmPylRS–*${\mathrm{tRNA}}_{\mathrm{CUA}}^{\mathrm{Pyl}}$ and pET-28–*sfGFP_R2TAG* were co-transformed into *E. coli* strain BL21(DE3). Single colonies were picked and pre-cultured in 1 mL of LB medium supplemented with Amp (100 μg/mL) and Kan (100 μg/mL) at 37 °C overnight. After cultivation, cells were harvested by centrifugation and resuspended in M9 minimal medium. In a ratio of 1:500, the pre-culture was transferred into 100 mL M9 minimal medium supplemented with Amp and Kan and grown at 37 °C until OD_600_ reached 0.6–0.8. Target gene expression was induced by the supplementation of 1 mM IPTG and ncAAs followed by incubation at 37 °C for 18 h. The cells were harvested and resuspended in phosphate buffered saline buffer (PBS buffer, pH 7.5). Lysozyme (0.5 mg/mL), DNase (10 µg/mL), and RNase (10 µg/mL) were added and the cells incubated at 4 °C for 30 min. Cells were further lysed via sonication (Sonopuls HD3200, Thermo Fisher Scientific, Germany) for 10 min (45% amplitude, 2 s pulse, 3 s pause). After centrifugation (45 min, 12,000 rcf, 4 °C), the supernatant was separated from the pellet. The supernatant was applied onto 1 mL Ni–NTA columns (GE Healthcare, München, Germany) for protein purification. First, 10 mL PBS buffer was used to wash the column. After supernatant application, non-target proteins were washed out by 10 mL washing buffer (1X PBS buffer, 20 mM imidazole, pH 7.4). Target proteins were eluted from the resin by 5 mL elution buffer (1X PBS buffer, 300 mM imidazole, pH 7.4). After purification, the buffer was exchanged to storage buffer (1X PBS buffer, pH 7.4) by dialysis.

### 4.5. Mass Spectrometry

To determine if *Mm*PylRS mutants mediate ncAA incorporation into sfGFP, intact protein masses were confirmed by electrospray ionization mass spectrometry (ESI-MS). After infusing the proteins through a C5 column (Supelco analytical, Sigma-Aldrich) by an Agilent 1260 HPLC system, ions were analyzed by an Agilent 6530 Q-TOF instrument. Spectra deconvolution was performed using Agilent Mass Hunter Qualitative Analysis software (v. B.06.00) employing the maximum entropy deconvolution algorithm. Observed protein masses were compared to theoretical values.

## 5. Conclusions

The class II aminoacyl–tRNA synthetase enzyme PylRS is structurally similar and belongs to the same evolutionary group as the bacterial phenylalanyl–tRNA synthetase (PheRS). Therefore, the change in substrate specificity of PylRS towards aromatic canonical and non-canonical amino acids by rational manipulation and library design is plausible. For example, the common evolutionary origin also explains why the Y384F mutants of PylRS can have better activity towards phenylalanine or tryptophan (and their analogs) [41,61,62].

In this study, we engineered the *Mm*PylRS synthetase towards the incorporation of several aromatic ncAA substrates. Over multiple stages, we selected positions in the enzyme for mutagenesis and used a fluorescence-based assay to screen for aaRS activity in vivo. Seven positions (L305, Y306, L309, N346, C348, V401, and W417) were chosen from the active site pocket of the aaRS. We found that modification of *Mm*PylRS at residue V401 significantly improved the enzyme performance with the sulfur-containing aromatic ncAA Bta. In addition, residue W417 expanded the substrate range of the enzyme towards Tyr analogs. The interest in site-specifically installed ncAAs has fueled continuous efforts to create and optimize OTSs. Even for the commonly catalytically superior *Mj*TyrRS systems, enzyme and host strain engineering efforts continue to improve target production and amber suppression background [63]. For PylRS, the accumulated literature allowed us to test if previously reported mutations also act on the Bta-specific enzyme created herein. Compiled from previous works, we selected several spatially distant positions (T13, V31, I36, T56, R61, H62, H63, A100, and S193), which are not directly involved in the binding pocket of the aaRS, for site-directed mutagenesis studies to improve efficiency. Interestingly, we found that synthetase performance in vivo benefits from mutation of residue S193, which is located in the C-terminal domain suggested to interact with tRNA^Pyl^. Therefore, our approach not only expands the *Mm*PylRS substrate range but also significantly improves the efficiency of the enzyme variants. Specifically, the incorporation of the well-known translationally active ncAA Bta proved to be enhanced. In addition, we successfully incorporated AzAla and 3-MeY site-specifically into proteins using a *Mm*PylRS–based OTSs for the first time. Again, this seems to be possible due to the enhanced activity of the orthogonal pair in *E. coli*.

Finally, we envision our approach as a strategy for overcoming the problems with the low efficiency of aaRS variants as a route to the discovery of new orthogonal enzymes. It focuses on the combined mutagenesis of first-shell and spatially distant residues, the latter for example belonging to the tRNA binding domain [34,36,37]. In this light, combination with computational approaches should lead to aaRS variants with specificities towards synthetic substrates and activity levels comparable to the wild-type aaRS [63]. This will significantly expand the chemical space and scope of orthogonal translation with numerous aromatic ncAAs useful in biophysics, cell and synthetic biology [53,64].

## Figures and Tables

**Figure 1 molecules-25-04418-f001:**
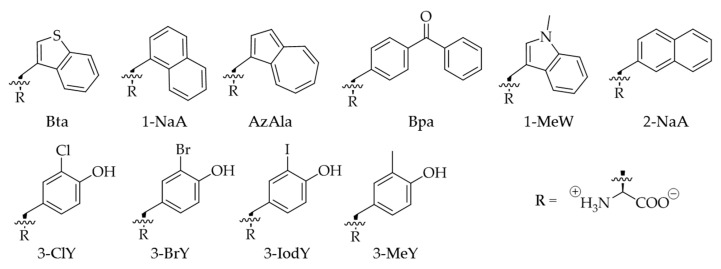
Chemical structure of Trp and Tyr analogs used in this study. 3-benzothienyl-l-alanine (Bta), 3-(1-naphthyl)-l-alanine (1-NaA), β-(1-azulenyl)-l-alanine (AzAla), 4-benzoyl-l-phenylalanine (Bpa), 1-methyl-l-tryptophan (1-MeW), 3-(2-naphthyl)-l-alanine (2-NaA), 3-chloro-l-tyrosine (3-ClY), 3-bromo-l-tyrosine (3-BrY), 3-iodo-l-tyrosine (3-IodY), and 3-methyl-l-tyrosine (3-MeY).

**Figure 2 molecules-25-04418-f002:**
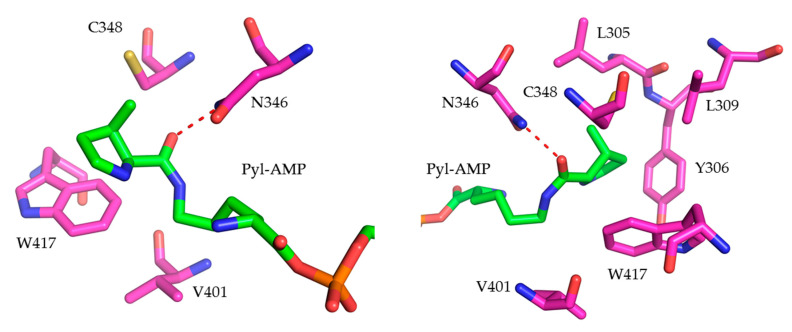
Key residues surrounding the active site pocket of *Mm*PylRS for amino acid substrate recognition (PDB: 2Q7H [10]). The distance between N346 and the bound Pyl–AMP as indicated (red dashed line) is 2.7 Å.

**Figure 3 molecules-25-04418-f003:**
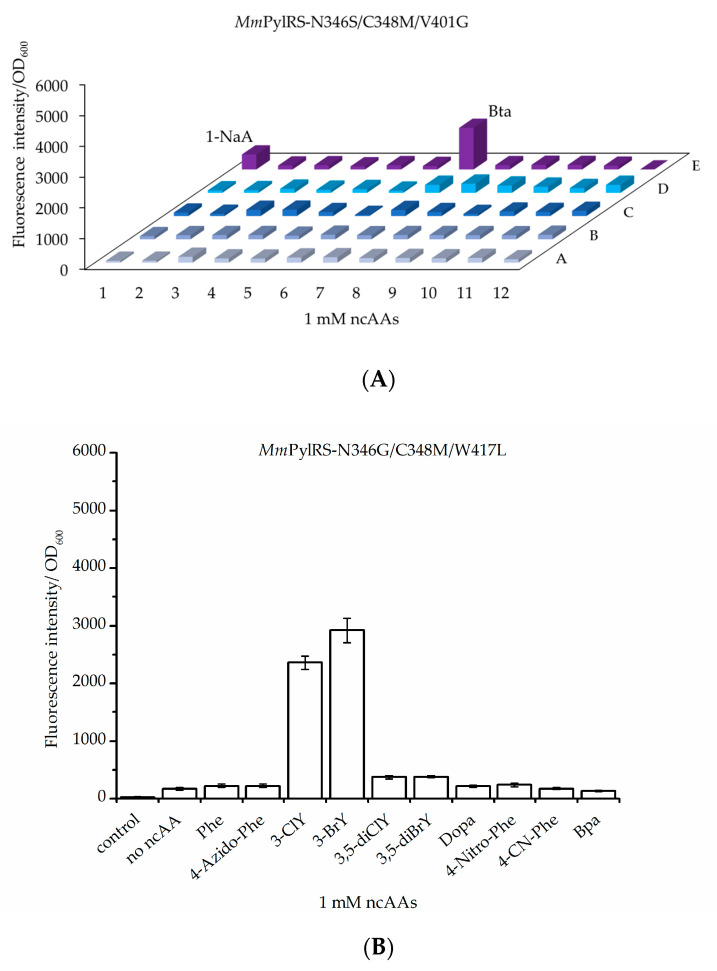
In vivo amber suppression capacities and analytics of ncAA incorporation at amber sites with two different *Mm*PylRS variants: (**A**) *Mm*PylRS–SMG and (**B**) *Mm*PylRS–GML. The library for ncAA incorporation was screened by monitoring sfGFP–R2TAG reporter fluorescence intensity. Details of the library screening with *Mm*PylRS variants and ncAAs in 96-well plates are given in Appendix A and S3. Each amino acid was supplied at a final concentration of 1 mM. (Control: no inducer and no ncAAs; no ncAAs: with inducer but without amino acids supplementation). Data in (**B**) is means ± SD (*n* = 3).

**Figure 4 molecules-25-04418-f004:**
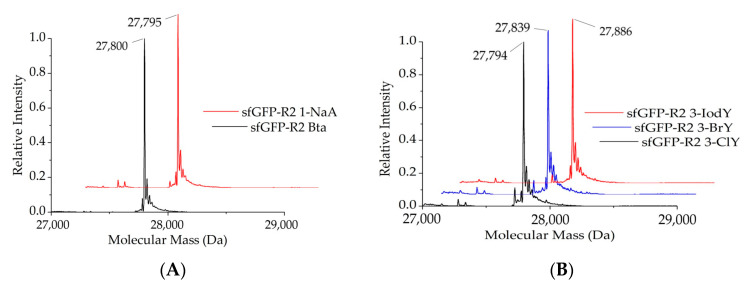
Deconvoluted ESI-MS profiles of the sfGFP–R2TAG reporter, revealing successful protein labelling with Trp and Tyr analogs. (**A**) Full-length sfGFP–R2_Bta and sfGFP–R2_1-NaA were produced by co-expression of *Mm*PylRS–SMG. The calculated molecular mass of sfGFP–R2_Bta is 27,800.31 Da, whereas the observed mass is 27,801 Da. The calculated molecular mass of sfGFP–R2_1-NaA is 27,794.3 Da whereas the observed mass is 27,794 Da. (**B**) Full-length sfGFP–R2_3-ClY, sfGFP–R2_3-BrY, and sfGFP–R2_3-IodY were produced by co-expression of *Mm*PylRS–GML. The calculated molecular mass of sfGFP–R2_3-ClY is 27,794.07 Da, whereas the observed mass is 27,795 Da. The calculated molecular mass of sfGFP–R2_3-BrY is 27,838.02 Da, whereas the observed mass is 27,839 Da. The calculated molecular mass of sfGFP–R2_3-IodY is 27,886 Da, whereas the observed mass is 27,886 Da.

**Figure 5 molecules-25-04418-f005:**
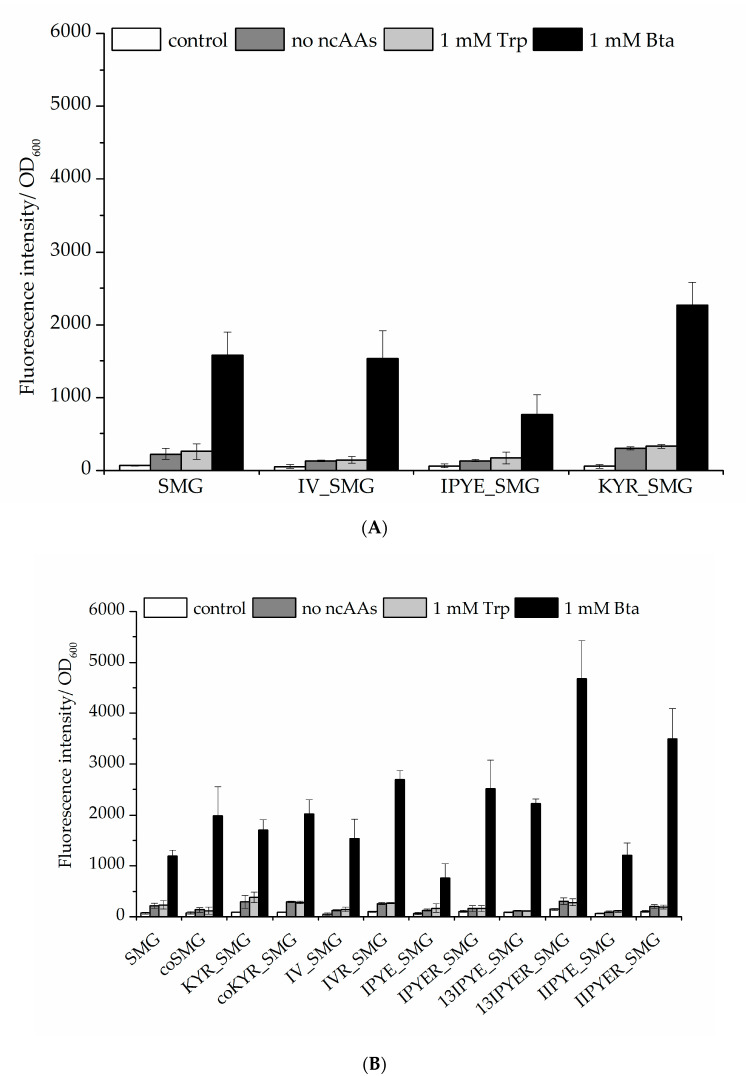
Comparison of Bta incorporation efficiency. Fluorescence of the sfGFP–R2TAG reporter construct arises from amber suppression mediated by different *Mm*PylRS–SMG variants. (**A**) *Mm*PylRS–SMG variants combined with activity improving mutation sets introduced by site-directed mutagenesis. According to the observed total cell fluorescence, variant *Mm*PylRS–KYR_SMG led to the production of more full-length sfGFP compared to other aaRS constructs. (**B**) *Mm*PylRS–SMG variant performance when combined with different sets of efficiency-improving mutations. Ideally, aaRS construct expression should lead to a low background signal in the absence of ncAA supplementation (no ncAAs controls) and high efficiency Bta incorporation (observed by fluorescence intensity) when the ncAA is supplied. The control setup is without induction and ncAA supplementation; the no-ncAAs setup is with IPTG induction but without Trp or Bta supplementation, respectively. Data are means ± SD (*n* = 3).

**Figure 6 molecules-25-04418-f006:**
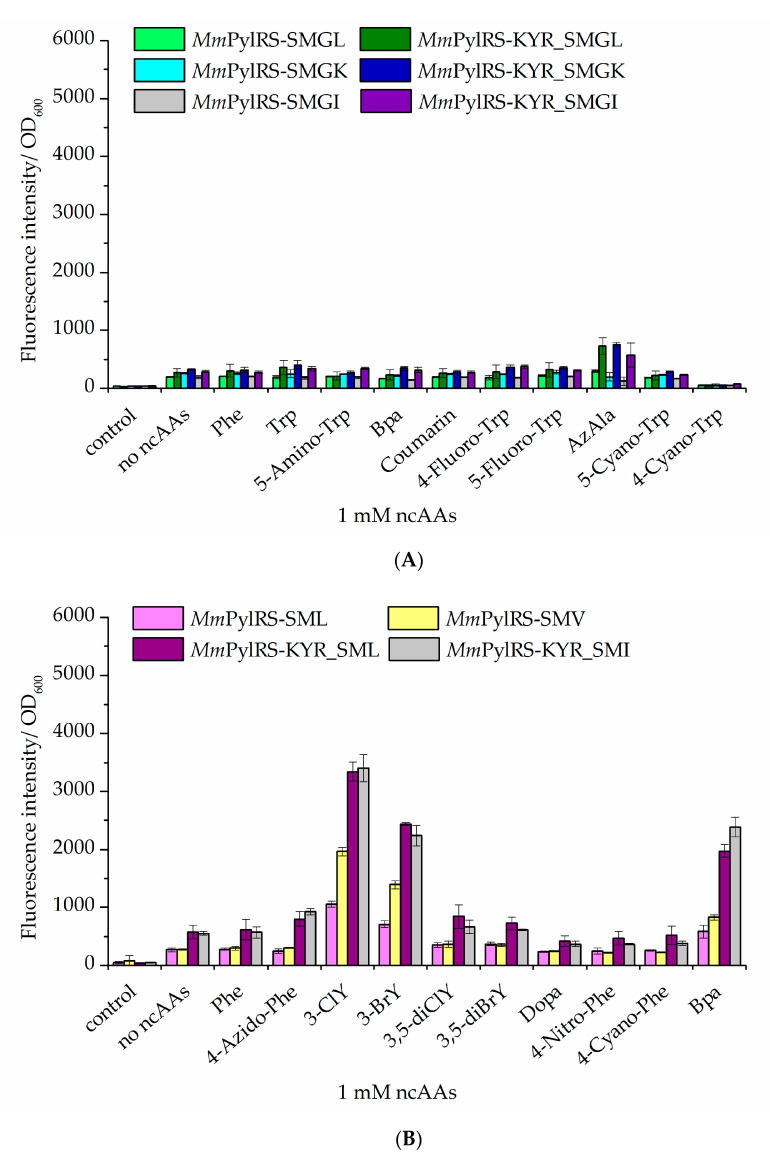
Comparison of ncAA incorporation via different *Mm*PylRS variants and impact of the activity-enhancing KYR mutation set (R61K, H63Y, and S193R). Fluorescence of the sfGFP–R2TAG reporter construct arises from amber suppression mediated by improved *Mm*PylRS–SMG variants. (**A**) Trp analogs were incorporated into sfGFP by different *Mm*PylRS variants (the last letter always indicating the substitution of aaRS position W417, mutation details in Appendix A; Control: without inducer and ncAAs). (**B**) Tyr analogs were incorporated into sfGFP by different *Mm*PylRS variants (control: without inducer and ncAAs; the no-ncAAs setup is with IPTG induction but without Trp or Bta supplementation, respectively). In both panels, data are means ± SD (*n* = 3).

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
