# Peer review of "Expanding the Scope of Orthogonal Translation with Pyrrolysyl-tRNA Synthetases Dedicated to Aromatic Amino Acids"

_molecules, 2020, doi:10.3390/molecules25194418_

Round 1

Reviewer 1 Report

In the manuscript titled ‘Expanding the scope of orthogonal translation with pyrrolysyl-tRNA synthetases dedicated to aromatic amino acids’ by Tseng et al. the authors present the results of several rounds of rational protein engineering to improve the performance of the Mm pyrrolysyl OTS with various aromatic ncAAs. This reviewer finds the work solid and has no major issues with either the methodology or the results. Some additional experiments/controls would improve the manuscript, but they are not required for publication. See below for specific comments.

Comments:

  1. I found the nomenclature used to define the mutants hard to follow given the letter combinations frequently referred to different positions, eg. SMG vs. GML. Given the number of positions tested, I understand there may not be an easy way to quickly convey this information, but the descriptions used here were confusing.

  1. As the authors correctly point out, a common problem with OTSs is very low activity. It would have been nice to see a control included in the fluorescence assays for sfGFP without a TAG codon and/or sfGFP-R2TAG with a suppressor tRNA as points of reference.

  1. In Figure 3 and Figure 6 the authors use Phe as the canonical amino acid control. Tyr would have been a better choice given the authors described the performance with Tyrosine analogs. Data in S4D shows minimal incorporation of either Phe or Tyr by GML.

  1. A useful addition would be an experiment showing the performance of the best PylRS variants in rich-defined or complex media. This would better represent many real applications of OTSs.

  1. Line 184. “Given that the suppressor tRNA abundances are not limiting, increasing the intracellular amount of soluble and functional synthetase will lead to increased full-length target protein yields. Comparing with the previous work, the codon-optimized MmPylRS increased the Bta incorporation efficiency (Figure S4).”

Are the authors not surprised that increasing the amount of MmPylRS present in the cell via codon optimizing the gene did not increase (and reportedly decreased) the background incorporation in the absence of a ncAA?

  1. Line 324. Correct typo. PlyRS.

  1. In S9 the authors state “Background signal: sfGFP with Glu incorporation is 27725 Da.

Should this be Gln? Glutamine is normally the strongest near-cognate suppressor, and I am not sure what the rationale is for assuming Glu would be present at residue 2.

Author Response

Reviewer 1: In the manuscript titled ‘Expanding the scope of orthogonal translation with pyrrolysyl-tRNA synthetases dedicated to aromatic amino acids’ by Tseng et al. the authors present the results of several rounds of rational protein engineering to improve the performance of the Mm pyrrolysyl OTS with various aromatic ncAAs. This reviewer finds the work solid and has no major issues with either the methodology or the results. Some additional experiments/controls would improve the manuscript, but they are not required for publication. See below for specific comments.

Reviewer 1 Question 1): I found the nomenclature used to define the mutants hard to follow given the letter combinations frequently referred to different positions, e.g. SMG vs. GML. Given the number of positions tested, I understand there may not be an easy way to quickly convey this information, but the descriptions used here were confusing.

Answer Reviewer 1 Question 1): We thank to the referee for this comment. Please see Supplementary Table 1 of our revised manuscript for mutations in the individual enzyme variants (such as MmPylRS-SMG which refers to MmPylRS (N346S:C348M:V401G). We now included additional references to the mutation scheme and a fitting Abbreviations section.

Reviewer 1 Question 2): As the authors correctly point out, a common problem with OTSs is very low activity. It would have been nice to see a control included in the fluorescence assays for sfGFP without a TAG codon and/or sfGFP-R2TAG with a suppressor tRNA as points of reference.

Answer Reviewer 1 Question 2): We appreciate the concern raised by the referee that points out an important issue. We performed the suggested experiments by using wild-type sfGFP (without a TAG codon, wt-sfGFP) and sfGFP-R2TAG with cognate suppressor tRNA, but without PylRS co-expression. They were added to fluorescence assays as further controls (Supplementary Figure 10). As expected, PylRS-based Bta incorporation into sfGFP-R2TAG, leads to lower target protein production and thus lower fluorescence intensities. Even though the OTS has lower activity than the endogenous ribosomal translation apparatus alone (in absence of amber suppression), it is evident that variant MmPylRS-KYR_SMG showed the highest Bta incorporation efficiency. This synthetase was optimized through our semi-rational design and screening strategy. The revised contexts were added in line 298-302 of this manuscript.

Reviewer 1 Question 3): In Figure 3 and Figure 6 the authors use Phe as the canonical amino acid control. Tyr would have been a better choice given the authors described the performance with Tyrosine analogs. Data in S4D shows minimal incorporation of either Phe or Tyr by GML.

Answer Reviewer 1 Question 3): We thank the referee for bringing this issue to our attention. Our choice to test Phe is based on the evolutionary connection of PylRS and PheRS (also discussed in our manuscript). On the other hand, the similarity between Phe and Tyr as well as the similarities in the recognition of hydrophobic amino acids are known in the field. We feel that the conducted MS analysis of produced target protein and the added experiments for production in (cAA-rich) complex media support this notion. Please note that the screening panels also include all cAAs individually added to chemically defined medium.

Reviewer 1 Question 4): A useful addition would be an experiment showing the performance of the best PylRS variants in rich-defined or complex media. This would better represent many real applications of OTSs.

Answer Reviewer 1 Question 4): We appreciate this relevant comment. Our work now includes a comparison of the wt-sfGFP expression and amber suppression by Bta incorporation in different growth media. As expected for the rich LB medium, target protein production was generally elevated compared to M9 medium (Supplementary Figures 11). Notably, orthogonal translation with MmPylRS-KYR_SMG displayed the highest Bta incorporation efficiency in LB medium. To corroborate these results, sfGFP proteins were purified and subjected to MS analysis. The results demonstrated that the chosen Bta-specific aaRS variants mediate Bta incorporation into the target protein in rich-defined and LB medium, as in M9 medium (Supplementary Figures 12). Therefore, the generated aaRS variants from our study can be utilized in variety of protein production scenarios and applications for ncAA-modified proteins. The revised contexts were added in line 303-312 of this manuscript.

Reviewer 1 Question 5): Line 184. “Given that the suppressor tRNA abundances are not limiting, increasing the intracellular amount of soluble and functional synthetase will lead to increased full-length target protein yields. Comparing with the previous work, the codon-optimized MmPylRS increased the Bta incorporation efficiency (Figure S4).”

Are the authors not surprised that increasing the amount of MmPylRS present in the cell via codon optimizing the gene did not increase (and reportedly decreased) the background incorporation in the absence of a ncAA?

Answer Reviewer 1 Question 5): We appreciate this relevant comment and agree this issue was not thoroughly explained in the original manuscript. Therefore, in the revised manuscript we have changed the wording as follows: ”Codon-optimizing the aaRS gene can enhance the translation efficiency and efficiency level within the bacterial host cell. As reported earlier for MbPylRS, it can increase the intracellular amount of soluble and functional synthetase, which in turn can increase full-length target protein yields. When compared with the natural archaeal gene sequence as used in previous works, the codon-optimized MmPylRS gene lead to an increased Bta incorporation efficiency (Figure S4)”. These sentences were added in line 201-206 of this manuscript. The small decrease in the no ncAA control is close to the region of experimental error, which is why we decided not speculate about its cause.

Reviewer 1 Question 6): Line 324. Correct typo. PlyRS.

Answer Reviewer 1 Question 6): Typos were removed in the revised manuscript.

Reviewer 1 Question 7): In S9 the author’s state “Background signal: sfGFP with Glu incorporation is 27725 Da.” Should this be Gln? Glutamine is normally the strongest near-cognate suppressor, and I am not sure what the rationale is for assuming Glu would be present at residue 2.

Answer Reviewer 1 Question 7): We thank to the referee for bringing this issue to our attention. We changed Glu to Gln and explained the background signal in more detail in the revised supporting information manuscript (Supplementary Figures 9).

Reviewer 2 Report

In this paper, Tseng, et al. expanded the substrate scope of Methanosarcina mazei pyrrolysyl tRNA synthetase (MmPylRS) to be able to incorporate aromatic amino acid residues. MmPylRS has been widely explored to incorporate various aliphatic pyrrolysine analogs, but was less studied to incorporate aromatic residues.  Tseng, et al. performed semi-rational design by randomizing the key residues in the MmPylRS substrate binding pocket that are important for substrate recognition, and performed high-throughput fluorescence-based screening to select MmPylRS variants that can incorporate tryptophan analogs. The incorporation of the noncanonical amino acids (ncAA) are further confirmed by mass-spec analysis of the sfGFP with amber-suppressed codon. Furthermore, they also identified a mutation, S193R, which can improve the general catalytic efficiency of MmPylRS. This new orthogonal system for incorporating ncAA is interesting because it provides alternative system for incorporating ncAA with aromatic side chains. I have the following suggestions to improve the manuscript:

  1. Currently it is not very obvious what is the advantage of this system over the existing MjTyrRS-based system for incorporating ncAA with aromatic side chains. It is best to discuss the advantages or comparison of the systems in more details regarding efficiency, substrate scope, organisms to be used, etc.
  2. In Figure 1, it is best to add a bond connecting the wedge and a scribbly line to indicate the connectivity. Otherwise, it is not obvious whether the aromatic sidechains are connected to the end of the wedge or the chiral Carbon.
  3. In Figure 2, it is best to indicate the distance rather than just showing the possible interactions by dashed lines
  4. The logic between Line 55-56 is not clear.
  5. There is probably some deletion of words in the sentence in line 210-211.

Author Response

Reviewer 2: In this paper, Tseng, et al. expanded the substrate scope of Methanosarcina mazei pyrrolysyl tRNA synthetase (MmPylRS) to be able to incorporate aromatic amino acid residues. MmPylRS has been widely explored to incorporate various aliphatic pyrrolysine analogs, but was less studied to incorporate aromatic residues.  Tseng, et al. performed semi-rational design by randomizing the key residues in the MmPylRS substrate binding pocket that are important for substrate recognition, and performed high-throughput fluorescence-based screening to select MmPylRS variants that can incorporate tryptophan analogs. The incorporation of the noncanonical amino acids (ncAA) are further confirmed by mass-spec analysis of the sfGFP with amber-suppressed codon. Furthermore, they also identified a mutation, S193R, which can improve the general catalytic efficiency of MmPylRS. This new orthogonal system for incorporating ncAA is interesting because it provides alternative system for incorporating ncAA with aromatic side chains. I have the following suggestions to improve the manuscript:

Reviewer 2 Question 1): Currently it is not very obvious what is the advantage of this system over the existing MjTyrRS-based system for incorporating ncAA with aromatic side chains. It is best to discuss the advantages or comparison of the systems in more details regarding efficiency, substrate scope, organisms to be used, etc.

Answer Reviewer 2 Question 1): Orthogonal pairs based on both MjTyrRS and Mm/MbPylRS scaffolds can have advantages and disadvantages in different microbial hosts. For example, in bacterial host cells, MjTyrRS often show a higher performance compared with PylRS-based systems. In general, orthogonal translation with PylRS-derived o-pairs leads to lower yields of target protein, since this enzyme generally has a low catalytic efficiency. While o-pairs based on MjTyrRS allow a higher number of in-frame stop codons that can be suppressed, this is not the case with orthogonal translation based on PylRS. However, the PylRS system particularly covers a wide range of applications for genetically encoding ncAAs with bio-orthogonality in both prokaryotes and eukaryotes. We feel that this point was introduced (Introduction section from line 74) and discussed (Discussion section from line 340). Our additional experiments for amber suppression efficiency compared to wt-sfGFP and the system performance in different cultivation media further show the advantage of our system.

Reviewer 2 Question 2: In Figure 1, it is best to add a bond connecting the wedge and a scribbly line to indicate the connectivity. Otherwise, it is not obvious whether the aromatic sidechains are connected to the end of the wedge or the chiral Carbon.

Answer Reviewer 2 Question 2: In the revised manuscript the old illustration version is replaced by a new one with scribble lines to indicate connectivity.

Reviewer 2 Question 3: In Figure 2, it is best to indicate the distance rather than just showing the possible interactions by dashed lines

Answer Reviewer 2 Question 3: As requested by the reviewer, we have now indicated the distance for the possible interaction.

Reviewer 2 Question 4: The logic between Line 55-56 is not clear.

Answer Reviewer 2 Question 4: We appreciate this referee comment. In the revised manuscript we changed this sentence towards more clarity.

Reviewer 2 Question 5: There is probably some deletion of words in the sentence in line 210-211.

Answer Reviewer 2 Question 5: We thank to the referee for bringing this to our attention. The sentence was changed in the revised manuscript.
